# Multifunctional PLA/Gelatin Bionanocomposites for Tailored Drug Delivery Systems

**DOI:** 10.3390/pharmaceutics14061138

**Published:** 2022-05-27

**Authors:** Carmen Moya-Lopez, Alberto Juan, Murillo Donizeti, Jesus Valcarcel, José A. Vazquez, Eduardo Solano, David Chapron, Patrice Bourson, Ivan Bravo, Carlos Alonso-Moreno, Pilar Clemente-Casares, Carlos Gracia-Fernández, Alessandro Longo, Georges Salloum-Abou-Jaoude, Alberto Ocaña, Manuel M. Piñeiro, Carolina Hermida-Merino, Daniel Hermida-Merino

**Affiliations:** 1Laboratoire Matériaux Optiques Photonique et Systèmes (LMOPS), Centrale Supélec, Université de Lorraine, 57000 Metz, France; carmen.moya-lopez-pelaez@univ-lorraine.fr (C.M.-L.); murillo.donizeti@ufabc.edu.br (M.D.); david.chapron@univ-lorraine.fr (D.C.); patrice.bourson@univ-lorraine.fr (P.B.); 2Centro Regional de Investigaciones Biomédicas, Unidad NanoCRIB, 02008 Albacete, Spain; alberto.juan@uclm.es (A.J.); ivan.bravo@uclm.es (I.B.); carlos.amoreno@uclm.es (C.A.-M.); 3Group of Recycling and Valorization of Waste Materials (REVAL), Marine Research Institute (IIM-CSIC), 36208 Vigo, Spain; jvalcarcel@iim.csic.es (J.V.); jvazquez@iim.csic.es (J.A.V.); 4NCD-SWEET Beamline, ALBA Synchrotron Light Source, 08290 Cerdanyola del Vallès, Spain; esolano@cells.es; 5Facultad de Farmacia de Albacete, Universidad de Castilla-La Mancha, 02008 Albacete, Spain; pilar.ccasares@uclm.es; 6Unidad de Medicina Molecular, Centro Regional de Investigaciones Biomédicas, 02008 Albacete, Spain; 7TA Instruments Waters Chromatography, Tres Cantos, 28760 Madrid, Spain; carlos_gracia@waters.com; 8ID20, ESRF, 71 Avenue des Martyrs, 38000 Grenoble, France; alessandro.longo@esrf.fr; 9Istituto per lo Studio dei Materiali Nanostrutturati (ISMN)-CNR, UOS Palermo, Via Ugo La Malfa, 153, 90146 Palermo, Italy; 10Constellium C-TEC Technology Center, Parc Economique Centr’alp, 725 Rue Aristide Bergès CS10027, 38341 Voreppe, France; georges.salloum-abou-jaoude@constellium.com; 11Experimental Therapeutics Unit, Hospital Clínico San Carlos, IdISSC and CIBERONC, 28040 Madrid, Spain; albertoo@sescam.jccm.es; 12Unidad de Investigación del Complejo Hospitalario Universitario de Albacete, Oncología Traslacional, 02008 Albacete, Spain; 13Departamento de Física Aplicada, CINBIO, Universidade de Vigo, Campus Lagoas-Marcosende, 36310 Vigo, Spain; mmpineiro@uvigo.es; 14Netherlands Organisation for Scientific Research (NWO), DUBBLE@ESRF BP CS40220, 38043 Grenoble, France

**Keywords:** bionanocomposite, gelatin, polylactide, stereocomplex, hydrogel, nanoparticles, drug delivery

## Abstract

A series of bionanocomposites composed of shark gelatin hydrogels and PLA nanoparticles featuring different nanostructures were designed to generate multifunctional drug delivery systems with tailored release rates required for personalized treatment approaches. The global conception of the systems was considered from the desired customization of the drug release while featuring the viscoelastic properties needed for their ease of storage and posterior local administration as well as their biocompatibility and cell growth capability for the successful administration at the biomolecular level. The hydrogel matrix offers the support to develop a direct thermal method to convert the typical kinetic trapped nanostructures afforded by the formulation method whilst avoiding the detrimental nanoparticle agglomeration that diminishes their therapeutic effect. The nanoparticles generated were successfully formulated with two different antitumoral compounds (doxorubicin and dasatinib) possessing different structures to prove the loading versatility of the drug delivery system. The bionanocomposites were characterized by several techniques (SEM, DLS, RAMAN, DSC, SAXS/WAXS and rheology) as well as their reversible sol–gel transition upon thermal treatment that occurs during the drug delivery system preparation and the thermal annealing step. In addition, the local applicability of the drug delivery system was assessed by the so-called “syringe test” to validate both the storage capability and its flow properties at simulated physiological conditions. Finally, the drug release profiles of the doxorubicin from both the PLA nanoparticles or the bionanocomposites were analyzed and correlated to the nanostructure of the drug delivery system.

## 1. Introduction

Nanoparticles (NP) emerged in the 1970s as a drug delivery system (DDS) approach to improve the stability and solubility of the encapsulated molecules, promote drug delivery across cellular membranes as well as enhance drug circulation duration, improving safety and efficacy [1]. The composition of the designed nanoparticles has ranged from natural materials such as chitosan, dextran and gelatin or inorganic materials such as gold, silica and Cd/Zn as well as synthetic polymeric nanoparticles [2]. In particular, polymeric nanoparticles have recently received great interest due to the possibility to tailor the drug release rate by varying polymeric architectural parameters such as crystallinity, hydrophilicity or glass transition temperature [3]. Polylactide (PLA) is among the most commonly used polymers for nanoparticle formulation development due to its biocompatibility. PLA is a bio-based and polymorphic polyester comprising a chiral carbon in its structure that yields a wide variety of nanostructured derivatives (isotactic, atactic, syndiotactic, stereo-complexed, etc.) [4,5,6] featuring different physicochemical properties. In addition, the processing conditions during the formation of the nanoparticles determine the kinetic trapped or thermodynamic state of the final material that affect significantly the final polymer characteristics such as crystallinity degree and crystalline phase, which influence the drug release rate [3]. Particularly, polyesters exhibiting low glass transition temperature (T_g_), melting temperature (T_m_), and degree of crystallinity generally feature higher drug release [7,8]. Furthermore, amorphous PLA is characterized by soft and elastic mechanical properties, whilst semicrystalline PLA is stiff and robust [9], which will invariably define the drug release. Likewise, the drug release rate from PLA nanoparticles exhibiting stereocomplex (SC) phase is expected to decrease compared to the homopolymer counterpart due to the stabilization by specific C-H…O-H hydrogen bonds in the crystalline phase that increases its hydrolytic resistance [10]. However, the impact of the processing parameters of the formation of the nanoparticles on the final nanoparticles release kinetics is found inconsistent due to the lack of mechanistic process control as well as the unusual analysis of the structure–properties relationship of the formulated nanoparticles. However, systematic studies correlating the structure-processing-properties relationship of nanoparticles formulation have been recently accomplished [3,11]. 

The hydrophobic character of PLA increases the contact angle with water, which diminishes the interaction with physiological fluids and cell adhesion, which is commonly faced by copolymerizing PLA with hydrophilic monomers such as polyethyleneglycol (PEG) or poly-glycolic acid (PGA) to afford block copolymers with enhanced biocompatibility [12,13,14]. Recently, nanoparticles generated from poly-L-lactide(PLLA)-3-ethylglycolide and poly-D-lactide (PDLA)-3-ethylglycolide were successfully formulated featuring the SC phase as well as attaining crystallinity degrees up to 50%. However, a long (7 days) stereocomplexation process from solution was required to obtain crystallized nanoparticles [15]. In addition, PLA micelles crystallized in stereocomplex phase were also achieved from PEG-PLLA-PDLA stereoblock copolymers by the nanoprecipitation method, which exhibited a slower drug release rate than the amorphous and isotactic counterpart [16]. However, the copolymerization of PLA with PGA or PEG usually deters the crystallization of the nanoparticles, limiting the physicochemical properties of the PLA-derivatives to the characteristic features of the amorphous phase [15] that neutralize the PLA chirality effect on the molecular mobility and the relaxation process [17]. Alternative processing methods are urged to favor the hydrophilicity of the PLA-based material whilst retaining the tunability of the physicochemical properties for the nanostructured PLA derivatives. Several bio-based materials have been previously used for the fabrication of bioscaffolds such as collagen, chitosan, alginate, gelatin, silk, etc. [18]. Alginate is an attractive material for biomedical applications due to its structural similarities to polysaccharides; however, it is unable to meet concomitantly the different required parameters for tissue engineering applications such as the degradation degree, the mechanical properties and bioactivity [19]. Similarly, collagen-based scaffolds for the regeneration of tissues such as nerves, bone, cartilage or skin were also developed with the detrimental low solubility in water that turns gelatin into a better candidate for tissue engineering [20]. Gelatin is a natural biopolymer obtained from the denaturation of collagen [21] that preserves key bioactive functions of collagen for biomedical applications, such as cell adhesion and proliferation [22]. Moreover, gelatin is widely used in the food and cosmetic industries, and particularly, fish gelatin is becoming increasingly used for biomedical applications due to a likely less intense immune reaction in the human body than mammalian gelatin [23] as well as an approach to revalorize the by-products from fish industrial processes [24]. Moreover, gelatin can be processed in several formats depending on the intended applications, such as electrospun matrices for wound healing [25], microspheres for being loaded with a growth factor for bone regeneration [26], nanoparticles as nanocarriers for anticancer therapeutic agents [27], protein, gene and vaccines delivery [28], DDS for regenerative medicine or three-dimensional scaffolds for drug screening [29]. PLA/gelatin-based scaffolds were frequently used as a matrix for tissue engineering or drug delivery. Recently, tetracycline-loaded PLLA microparticles embedded in a porous gelatin scaffold exhibited controlled release behavior, biocompatibility as well as antibacterial activity [30]. Furthermore, an electrospun composite scaffold of polycaprolactone and gelatin with curcumin-loaded PLA microparticles exhibited biocompatibility and good mechanical properties for wound-healing applications [31]. However, the design of bionanocomposites containing crystalline PLA nanoparticles was generally dismissed as an approach to tailor the drug delivery rate.

Herein, a multifunctional biomatrix based on gelatin was used to generate a series of bionanocomposites with PLA nanoparticles exhibiting different crystal phases and crystallinities in an effortless way to afford drug vehicle systems with complex releasing profiles. PLA nanoparticles based on three different PLA derivatives were formulated with two different antitumoral compounds to personalize the drug release, namely, racemic-PLA (rac-PLA, amorphous), homopolymer-PLLA (HC-PLA, semicrystalline), and stereo-diblock-PLA (SC-PLA, semicrystalline). PLA nanoparticles were subsequently blended in an aqueous solution with fish gelatin obtained from waste fisheries at 65 °C to disperse them before annealing the polymeric component to achieve the aimed crystal phase and crystallinity degree. The nanoparticles generated were characterized by dynamic light scattering (DLS) and scanning electron microscopy (SEM). The thermal transitions of the gelatin/PLA bionanocomposites were studied by differential scanning calorimetry (DSC), and their nanostructures were probed by wide- and small-angle X-ray scattering (WAXS and SAXS) techniques as well as their structural development during both the sol–gel transition and the effect annealing time on the crystallinity and crystal phase of PLA nanoparticles. The storage capability of the drug delivery system and the feasibility of the local administration were assessed by rheological studies and in particular, the so-called “syringe test” at two different shear rates and temperatures to simulate physiological conditions. In addition, drug release studies from both the PLA nanoparticles, gelatin hydrogel network and their corresponding bionanocomposites have proved the impact of the gelatin network on the release kinetics. Likewise, the gelatin matrix was employed to provide to the drug vehicle system the mechanical properties to firstly store and then administer locally the therapeutic agent in a simple way by a syringe to lessen the harmful effect on healthy cells [32]. In addition, the gelatin hydrogel network acts as suitable support to both anneal the polymeric nanoparticles to customize their release profile whilst retaining the bioactive functions and to load compatible therapeutic agents with different release profiles for combined treatments.

## 2. Materials and Methods

Solvents and reagents were acquired from Sigma-Aldrich. Blue Shark (BS, *Prionace glauca*) skin by-products were kindly provided by Propegal S.L. L- and D-lactide dimers were purchased from Rex scientific. The lactide dimers were purified three times by sublimation and then stored in a glovebox at 4 °C. Toluene and tetrahydrofuran (THF) were pre-dried over sodium wire as well as distilled under nitrogen from sodium and subsequently stored over molecular sieves (3 Å) in a glovebox.

Gelatin extraction. For the production of gelatin (GE), frozen skins of BS were cut in portions of around 5 × 5 cm and, after defrosting, were washed with water for 30 min under orbital agitation to remove impurities. Then, sequential chemical treatments were applied to skins, in all cases using a (1:4) solid: liquid ratio and continuous agitation (50 rpm). Each chemical stage was run for 30 min at 22 °C, intercalating water washings in between treatments. The first treatment of skins was performed with 0.05 M NaOH, which was followed by acid processing using a solution of H_2_SO_4_ (0.02 M) and ending with another solution of citric acid (0.052 M). Thereafter, skins were soaked in water (1:2 ratio) at 45 °C for 16 h and gentle stirring to achieve the gelatin extraction in an aqueous medium. The obtained gelatin solution was then separated from skins by filtration (500 µm) and mixed with active charcoal (1.5%, *w*/*v*) for 3 h under constant agitation (100 rpm). Charcoal and impurities were removed by centrifugation (8000× *g*/20 min), and the gelatin was dried in an air-forced convection oven at 50 °C for 72 h. 

Polymerizations. Polymerizations were performed as described previously [33]. Briefly, two Schlenk tubes were loaded in the glovebox with the required amount of LA and the initiator and a magnetic stirrer, separately, and dissolved in toluene. Both LA and initiator Schlenk flasks were attached to the vacuum line, and solutions were poured together after temperature equilibrium at 90 °C was ensured. Stereo-diblock copolymerization was attained by adding the second monomer after the complete polymerization of the first monomer by monitoring the reaction by ^1^H NMR. Methanol was used to terminate the reaction and precipitate out the afforded polymer. The obtained polymer was collected by filtration, dried at room temperature, exposed to vacuum for over 24 h in the Schlenk line and stored upon characterization. Three PLA derivatives were obtained depending on the monomer used, namely, racemic PLA (rac-PLA), homopolymer PLA (HC-PLA) and stereo-diblock PLA (SC-PLA). 

Nanoparticles formulation. PLA nanoparticles (NP-PLA) were formulated from the three PLA derivatives, racemic-PLA (NP-rac-PLA), homopolymer-PLA (NP-HC-PLA) and stereo-diblock-PLA (NP-SC-PLA). In addition, all the PLA derivatives NP were loaded with the antitumoral doxorubicin and dasatinib as reference drugs [34,35,36]. Nanoparticles were prepared following the double emulsion methodology. Briefly, 10 mg of PLA was dissolved in 4 mL of CH_2_Cl_2_ and 1 mL of mQ water was added; then, the mixture was shaken in a Vortex and subsequently sonicated in an ice bath for 1 min to avoid emulsion overheating. The pre-emulsion generated was added to 10 mL of PVA (1%), shaken in a Vortex and sonicated for 5 min. The emulsion was allowed to evaporate organic solvent at room temperature for 45 min in a fume hood; then, it was centrifuged at 4 °C and 15 k rpm for 20 min. Nanoparticles were collected and dissolved in PBS pH = 7. Drug-encapsulated nanoparticles were prepared following the above-mentioned methodology. The amount of the drug (1 mg) was dissolved along with the polymer in 4 mL of CH_2_Cl_2_. The encapsulated drug in the NP-PLA is indicated at the end of the abbreviation name: NP-rac-PLA/DOX. The correspondence between the sample abbreviations and their compositions is summarized in the Appendix A.

Bionanocomposites. The bionanocomposites (GE/NP-Rac-PLA, GE/NP-HC-PLA, GE/NP-SC-PLA, GE/NP-Rac-PLA/DOX, GE/NP-HC-PLA/DOX, GE/NP-SC-PLA/DOX) were prepared by blending NP-PLA or NP-PLA/DOX with shark skin gelatin and tri-distilled water in a 0.4/25/74.6 *w*/*w* (NP/GE/H_2_O) proportion under constant stirring in an ultrasound bath (SELECTA, 40 kHz, 125 W) at 65 °C for 30 min. Furthermore, a hydrogel of gelatin (GE) was prepared at the same ratio 25/75 (GE/H_2_O) as a reference to compare the drug release [32]. Likewise, the corresponding sample abbreviation with its chemical composition is listed in the Appendix A. 

^1^H NMR. Spectra were recorded at room temperature on a Varian Inova FT-400/500 spectrometer and referenced to the standard in the deuterated solvent with the relaxation time fixed to 4 s. The polymeric derivatives were dissolved in CDCl_3_ for the synthetic characterization by ^1^H NMR. 

GPC measurements. The molecular weight and polydispersity index of the obtained PLA derivatives were determined by gel permeation chromatography (GPC) with a Shimadzu Prominence-I LC-2030 equipped with a Shodex GPC KF-805L column (Shodex, Tokyo, Japan) and a Shimadzu RID-20A detector (Shimadzu, Kyoto, Japan). CHCl_3_ was used as the mobile phase at 40 °C, with 1mL/min as the flow rate. Then, 5 mg of the PLA derivatives was dissolved in 1.5 mL of solvent overnight in constant agitation. Subsequently, the PLA solutions were filtered through a 0.2 µm polytetrafluoroethylene (PTFE) syringe filter before injection. Polystyrene with a molecular weight of 172 kDa was used as a reference to determine the overall molecular weight of the samples.

Optical rotation. The specific optical rotation [α] was measured by a JASCO P-20000 WI using a beam wavelength of 598 nm, using dichloromethane as solvent at a concentration of 1 mg/mL.

DLS technique. The nanoparticle size and its polydispersity index were measured in a 0.5% *v*/*v* PBS solution by using DLS techniques in a Zetasizer Nano ZS instrument (Malvern Instruments, Malvern, UK).

SEM. NP-PLA were freeze-dried and Au-shadowed before SEM images were recorded on a Jeol 6490LV electron microscope to evaluate the size and morphology of the particles. 

The resulting images were analyzed using Digital Micrograph™ software from Gatan. Moreover, NP-PLA were quantified using ImageJ software [37]. Filters were applied to the images to normalize the contrast and highlight the edges of the nanoparticles. Then, a suitable threshold was used to binarize the images. A Watershed binary image segmentation was applied to separate touching particles. Particle agglomerates were generated in most cases, since sample preparation was complicated, individual particles were considered for sample characterization/quantification using image analysis, and large particle agglomerates were avoided as much as possible. 

Cryo-SEM. The nanostructures of the GE/NP-HC-PLA, GE/NP-SC-PLA and GE/NP-Rac-PLA were evaluated in cryogenic mode. The hydrogels were mounted on the 13 mm gold sample holder and frozen at −200 °C with liquid nitrogen. Subsequently, the samples were transferred under liquid nitrogen in a cryogenic box to the Baltec equipment (MODEL, MED-020) and cut into 2–3 mm thickness slices. Finally, the samples were analyzed with an SEM (JEOL JSM-6700) at an acceleration voltage of 5 kV. 

Cryo-SEM images at a magnification of ×2000 were selected for the quantitative analysis due to the presence of a significant number of pores as statistically representative of the system. The coral-like porous structure in the micrographs was examined using ImageJ software [37]. Filters were applied in order to highlight the edges of the cells and be able to threshold and segment them. Several closing steps were applied also to clean artefacts in the images and only pores having an equivalent diameter bigger than 0.1 μm were quantified to plot the cell size distributions. 

DSC measurements. Differential scanning calorimetry (DSC) analyses were performed using a Q2000 DSC (TA Instruments, New Castle, DE, USA). The calibration was performed by using indium as a standard (T_m_ = 156.61 °C). DSC samples were prepared by sealing around 5 mg of the samples in Tzero Hermetic aluminum pans (TA Instruments, New Castle, DE, USA). The thermal protocol performed for bulk polymers consisted of a first heating ramp at 5 °C/min from 10 to 200 °C for Rac-PLA and HC-PLA, and from 25 to 220 °C for SC-PLA, followed by a cooling ramp at 5 °C/min to 10 °C and 25 °C, respectively. Moreover, the thermal protocol performed for NP-PLA and bionanocomposites (Gelatin with NP) consisted of a first heating ramp at 5 °C/min from 10 to 65 °C, followed by an isothermal step for 60 min. Samples were cooled down to 10 °C at 5 °C/min and further heated at 5 °C/min to 200 °C for the Rac and HC samples and to 220 °C for the SC samples. The thermal transitions (T_g_, T_c_, and T_m_) and their corresponding enthalpies (ΔHc, ΔHm) were calculated from the obtained thermograms. 

Raman spectroscopy. Raman measurements were performed with a micro-Raman spectrometer LabRAM HR from Horiba Jobin with a laser at 532 nm. The spectral resolution was around 1 cm^−1^ and a 50× microscope objective was used.

Small-angle X-ray scattering (SAXS) and wide-angle X-ray scattering (WAXS). SAXS/WAXS experiments were conducted both at DUBBLE (bm26) ESRF in Grenoble (France) and bl11 NCD-SWEET at ALBA, Cerdanyola del Vallès (Spain) with an X-ray wavelength of 12 Kev. The WAXS patterns were recorded with a Pilatus 300 kw (1472 × 195 pixels configuration) with a pixel size of 172 µm^2^ × 172 µm^2^ in DUBBLE, whereas a Rayonix LX255-HS characterized by a pixel size of 40 µm^2^ × 40 µm^2^ and active area of 85 mm^2^ × 255 mm^2^ (h × v) was used at NCD-SWEET. The SAXS patterns were collected with a Pilatus 1M with a detector size (981 × 1043) with a pixel size of 172 µm × 172 µm at a sample to detector distance of ca. 2.5 m. The calibration of the scattering angular was performed with either α-Al_2_O_3_ (alumina) for the WAXS scattering angles or AgBe for the SAXS scattering angle range. The measured intensity in the 2D detectors was reduced to 1D intensity profiles using bubble [21] as a function of the scattering vector (*q* = 4π/λsinθ) and reported in arbitrary units by correcting by the incident intensity and background subtraction.

The reduced SAXS experimental data of the hydrogel derivatives were adjusted to a heterogeneous sphere with fractal structure defined with a radius of gyration *Rg* related to the fractal size as well as the fractal dimension *D* that represent their compactness [38] by an inhouse program [39,40] connected to the minimization routine used by the MINUIT [41] program conceived in CERN, and thus, the SAXS profile intensity (*I*(*q*)) was fitted to:$$I(q)=\frac{\left(D-1\right)\sin\left[\left(D-1\right)\tan^{-1}\left(q,\xi \right)\right]}{\left(D-1\right)q\xi \;{\left(1+q^{2}\xi^{2}\right)}^{\frac{\left(D-1\right)}{2}}}$$
with
$$\xi^{2}=\frac{2R_{g}{}^{2}}{D\left(D+1\right)}$$ξ being the correlation length.

Rheological experiments. The mechanical properties were determined using a Physica MCR 101 rheometer (Anton Paar, Graz, Austria) able to control torques between 0.5 μN·m and 125 mN·m. Strain and sweep frequency sweeps for GE, GE/NP-Rac-PLA, GE/NP-SC-PLA and GE/NP-HC-PLA were performed using a cone-plate geometry (CP 25-1) with a constant space of 0.048 mm. The temperature was controlled using a Peltier P-PTD 200, which was placed at the lower plate [42]. The rheological experiments were repeated three times to analyze the reproducibility of obtained results, and the deviations in (complex) viscosities between repetitions were lower than 3%.

The linear viscoelastic range was previously determined by performing a strain sweep experiment in the range from 0.1 to 1000% at a constant angular frequency of 10 rad/s at 20 °C. The storage modulus G′ and the loss modulus G″ were determined in the linear strain range using a constant 0.1% strain. Frequency sweep measurements were conducted from 0.05 to 200 rad/s applying a constant strain of 0.1% at 20 °C. The solutions were preheated to 60 °C for 30 min for all rheological experiments to homogenize the samples and ensure adequate filling of the geometry as well as avoid air bubbles. 

The so-called “syringe test” experiments were performed using a plate–plate geometry (PP25/S) with a plate diameter of 25 mm and a geometric gap of 0.1 mm. A three-step experiment [43] was performed to evaluate the response of the gelatin-based bionanocomposites to mimic the injection in the human body of the DDS. Firstly, samples were subjected to a constant shear rate of 0.1 s^−1^ at room temperature, which simulates the resting state of the formulation under storage conditions and in the injection syringe. Subsequently, samples were heated up to 37 °C at a constant shear rate of 100 s^−1^, which simulates an injection using a needle with a diameter equal to 0.9 mm, which was typically selected for the administration of hydrogel formulations. Finally, a shear rate of 0.1 s^−1^ was imposed at 37 °C, simulating the rest state of the GE/NP-PLA hydrogel mixture in the human body.

Release studies. For NP-PLA/DOX, 10 mg of lyophilized DOX-loaded NPs were placed in the dialysis membrane (molecular weight cut-off: 3500 kDa) and incubated in 8 mL of phosphate-buffered saline (PBS, pH 7.4). The suspension was incubated at 37 °C with continuous stirring (50 rpm) in an IKA incubator shaker KS 3000. At different intervals of incubation, 1 mL of releasing medium was removed to measure the DOX absorbance in a UV-Vis Carey 100 spectrophotometer at 470 nm, and new PBS was added to maintain the volume constant.

For GE/NP-PLA/DOX, 1 mL of gelatin loaded with 5 mg of DOX-loaded NPs was placed in a dialysis membrane and released in the same method described above for the NP-PLA/DOX counterparts.

## 3. Results and Discussion

The incessant requirement by the pharmaceutic industry to develop novel drug delivery systems with enhanced therapeutic efficacy by maintaining the treatment drug concentration in the blood flow as well as maximizing the cell uptake has promoted the design of multiple drug delivery systems. Drug encapsulation permits drug stability during the transport before its release, which ideally must be directed to the targeted cells in an adequate treatment dose to minimize the adverse effects. In addition, the combination of different therapeutic agents with customized release rates must be involved in the treatment of tumors with typical post-surgical effects. The fulfillment of the performance requirements such as drug storage and administration are needed to meet multiple design requisites to attain the market. The nanomedicine approaches permit the conception of multifunctional materials with ambivalent properties that minimize the invasive detrimental effects of the therapeutic agents and their administration. Polymeric nanoparticles have been largely employed for drug encapsulation, with particular emphasis on PLA derivatives due to their biodegradability and abundance in the biocompatible polymer family. 

However, the clinical practice with PLA nanoparticles is still scarce, which might reflect the demanding features desired by DDS design to attain the clinical trials that were met exclusively at present by Genexol © PM (product based on PLA nanoparticles in phase III of clinical trials in the EU as well as in the US) [44]. Numerous studies were focused on the parameters that control the drug load/release processes by the polymeric nanoparticles, highlighting the nanostructure importance and in particular the crystallinity role.

In addition, the desired regulation of the pharmacokinetic and pharmacodynamic process of the industrial scale-up nanoparticle production involves the reproducibility formation conditions between different batches that determine the nanoparticle physicochemical features to allow their clinical translation [45]. Likewise, the tedious polymeric nanoparticle manufacturing that usually involves multiple synthetic steps compromises reproducibility and small yields [44]. Recently, microfluidics methodologies offer reproducible nanoparticle batches that enhance the viability of their application to the clinical step [46]. Likewise, the design of the polymeric nanoparticles should consider the therapeutic desired outcome of the clinical trial to reach the end product market due to the extensive investments needed that deserve a clear clinical as well as a profitable enhancement over the current treatment [47].

Likewise, interdisciplinary strategies are required to conceive novel formulations to meet the end-user needs of the generated PLA-based nanoparticles to attain the pharmaceutical clinical state. 

In particular, the catalyst employed in the PLA synthesis should be considered to limit the tin residue below 20 ppm to be applied for biomedical applications [48]. The development of catalyst systems is based on metals existing in the human body, such as metalloproteins with Zn or Mg. These Zn/Mg compounds are potential substitutes to limit the detrimental health effects of Sn-based catalysts, which are usually stocked in lungs and brain tissue.

Novel PLA derivatives with different nanostructures and medium molecular weight (M_n_) were generated, namely racemic-PLA (PDLLA; rac-PLA), homopolymer-PLA (PLLA; HC-PLA) and in particular the long-desired stereo-diblock-PLA (PLLA-*b*-PDLA; SC-PLA), by Ring-Opening Polymerization (ROP) using our novel alkyl zinc organometallic heteroscorpionate derivative as previously described [33] that minimizes the harmful effects. The selected PLA derivatives possess either amorphous or semicrystalline nanostructures with different crystalline phases. The molecular structure and M_n_ of the PLA derivatives were characterized by ^1^H NMR and GPC, respectively (Table 1 and Appendix A). Moreover, the L-LA/D-LA ratio was estimated to be higher than 98% for the HC-PLA and close to 50% for rac-PLA and SC-PLA, as analyzed by polarimetry measurements (Table 1). Furthermore, the P_m_ values, known as the probability of isotactic enchainment in which P_m_ = 0.5 is a random insertion whilst isotactic enchainment is designated by P_m_ = 1 [49], were obtained from deconvoluted homonuclear decoupling ^1^H NMR spectra (Appendix A) and confirmed the atactic character of the rac-PLA and the isotactic character of HC-PLA and SC-PLA (Table 1). 

Firstly, the nanostructure of the bulk polymer derivatives was characterized to understand the structure–properties relationship of the selected PLA derivatives to design different release rates. The thermal transitions of the bulk polymers were studied by DSC (Table 1) to understand the crystallization mechanism that will determine the nanostructure generated at the formulation conditions. Rac-PLA exhibited a T_g_ at 24 °C, which was notably lower than values reported in the literature [50], which is likely due to the lower molecular weight [51], whilst isotactic PLA-derivatives, namely HC- and SC-PLA, exhibited a T_g_ c.a. 40 °C. Additionally, only the HC- and SC-PLA exhibited crystallization and melting transitions, as both the T_c_ and T_m_ of SC-PLA were 13 °C higher than those of HC-PLA (Figure 1A). Notably, the HC-PLA exhibited a double endothermic peak at 161 °C (P_HC1_) and 167 °C (P_HC2_), corresponding to the recrystallization of the α’- into the α-phase and the subsequent melting of the latter [52]. Moreover, the SC-PLA exhibited also a double melting peak (at 180 °C (P_SC1_) and 195 °C (P_SC2_)) likely ascribed to the melt-recrystallization process of the α-phase into the stereocomplex (SC) phase [53]. The melting enthalpy for both HC- and SC- PLA was ca. 47 J/g. However, the crystallinity was 20% higher for the HC-PLA due to the lower ΔH_m_° value. 

Furthermore, the crystal phase of the bulk polymers was analyzed by Wide-Angle X-ray Scattering (WAXS) (Figure 1B). Only an amorphous halo was found for rac-PLA in agreement with the atactic character obtained from the P_m_ value. The characteristic pattern of the α-phase at (010), (110)/(200), (203) and (015) (q (nm^−1^)= 10.65, 11.9, 13.6 and 15.7) was observed for the HC-PLA, whilst the SC-PLA showed high-intensity reflections of the SC phase at (110), (300)/(030) and (220) (q (nm^−1^) = 8.5, 14.7 and 17)) and a low-intensity reflection at (110)/(200) characteristic of the α-phase. The presence of both α- and SC- phases on the SC-PLA is likely due to the typical crystallization competition occurring between homo- and stereocomplexation [53] when a quenching process occurs, such as the precipitation with methanol after the synthesis in agreement with the double-melting peak observed by DSC. 

In addition, Raman spectroscopy analysis of PLA bulk polymers (full spectrum shown in Figure 2A) was accomplished to identify the vibrational bands associated with phase variations (Figure 2B–E). In particular, the *v*C-COO stretching vibration band in the spectral range between 850 and 1000 cm^−1^ characterizes the helix conformation of the PLA polymer chain (Figure 2B). Likewise, the vibrational band assigned typically to *v*C-COO stretching at 870 cm^−1^ was found to shift to a higher wavenumber, to 880 cm^−1^, for SC-PLA [54]. Moreover, a second vibrational band of medium intensity, attributed to the coupling of the *v*C-C backbone stretching mode with the CH_3_ rocking mode, is observed either at 920 or 908 cm^−1^ for HC-PLA and SC-PLA, respectively, whilst it remained absent for the rac-PLA, reflecting that the PLA chain retains either the 10_3_ helix or the 3_1_ helix conformation for the corresponding α- and SC- phases (Figure 2B). Furthermore, the vibrational band associated with the stretching of the PLA carbonyl (C=O) appears mainly at ca. 1770 cm^−1^ for the racemic PLA derivative [55] (Figure 2C), confirming its amorphous nature, whereas the HC-PLA carbonyl spectral region splits in three overlapping bands at 1755, 1766 and 1777 cm^−1^, as reported previously for isotactic PLA (Figure 2D). In addition, the carbonyl band of SC-PLA appeared centered with a strong and sharp band at 1752 cm^−1^, which arose from the symmetric stretching vibrations [15], which are due to the stabilization of the stereocomplex crystalline phase by hydrogen bonds [10], and a broad shoulder at ca. 1770 cm^−1^ resulting from the vibration of the amorphous component (Figure 2E).

Polymeric nanoparticles from the selected PLA derivatives (NP-PLA) were successfully formulated with two different antitumoral compounds by the double emulsion method and their size and morphology were characterized by DLS (Table 2) and SEM (Figure 3 and Appendix A). Doxorubicin and dasatinib were selected to assess the loading capacity of the NP-PLA with two different model compounds that possess distinct structures either featuring a stiff structure imposed by the aromatic groups for the doxorubicin or a flexible chain structure for dasatinib. An averaged unimodal hydrodynamic diameter of 254 ± 22 nm with narrow PDI (0.1 ± 0.03) was obtained for all formulations by DLS. Furthermore, fairly spherical nanoparticles with a roundness index of 0.86 (being 1 perfectly round) and an average diameter of 104 ± 4nm were observed for the generated nanoparticle formulations by SEM. The size discrepancies observed by both techniques are related to shrinking effects during the freeze-drying and Au-shadowed treatment required for the SEM characterization, whilst DLS measurements were conducted in solution, resulting in NP swelling. In addition, the hydrodynamic diameter calculated by DLS arises from the diffusion coefficient of nanoparticles in suspension, in which small amounts of aggregates or dust are known to disturb the size determination [56,57]. Additionally, DLS analysis was performed at longer times to verify (PDI values remained constant with time) their stability in solution and thus the NP storage capability prior to their incorporation into the gelatin hydrogel network. In addition, the nanostructure of the formulated NP-PLA was characterized to understand the effect of the processing conditions on the PLA crystallization mechanism and its correlation with the release profiles.

The nanostructure of the lyophilized NP-PLA with doxorubicin (NP-PLA/DOX) probed by WAXS evidenced the kinetically trapped state in the amorphous phase for the PLA derivatives (Figure 4A) during the NP formation, removing the targeted crystallinity of the designed semicrystalline PLA derivatives (NP-HC-PLA and NP-SC-PLA). 

The amorphous structure obtained from the formulation conditions is likely related to the subjected short incubation time of the polymer solution during the water quenching conducted to generate the NP-PLA that prevents the polymer chains to nucleate to form the polymer crystallites [15]. However, the WAXS profiles of the NP-PLA/DOX exhibited crystalline reflections at high q values (q (nm^−1^) = 19.3, 22.4, 38.8 and 45, Figure 4B), corresponding to doxorubicin reflections [58], indicating that doxorubicin maintained its crystalline structure upon encapsulation in PLA nanoparticles. In addition, the NP-SC-PLA/DOX featured a crystalline structure (Figure 4A, zoom), which suggests the nucleating effect of doxorubicin. Likewise, the absence of the long period in the SAXS profile of the loaded nanoparticles confirms the quenching mechanism during the formation of the nanoparticles (Figure 4B). 

Thermal treatment for the formulated NP-PLA was applied by annealing them above the observed T_g_ at 40 °C (at 65 °C) for 1 h to modify the kinetically trapped nanostructure afforded upon formulation whilst selecting a temperature that will maintain the stability of the gelatin matrix. The NP-PLA T_g_ was followed by an endothermic peak at ca. 55 °C that might correspond to an enthalpy relaxation that occurred due to physical ageing during the storage period before the thermal analysis (Appendix A). Both NP-HC-PLA and NP-SC-PLA exhibited a cold crystallization during the isothermal annealing, confirming the quenched effect by the NP formation and discarding any degradation effect upon formulation (Figure 4C). NP-Rac-PLA did not crystallize upon heating in agreement with the bulk polymer structure. Additionally, upon further heating, NP-HC-PLA and NP-SC-PLA exhibited an endothermic peak at 165 °C and 200 °C, respectively, which is consistent with the melting of the α- and SC- phases, respectively, corroborating the crystallization of the semicrystalline PLA derivatives formulations during the annealing step (Figure 4D). Notably, an exothermal peak is observed prior to the melting peak of NP-HC-PLA that can be attributed to the recrystallization of the α’-phase into the α-phase [52], which is similar to the results obtained for the bulk polymer. Moreover, a double-melting peak was observed for the NP-SC-PLA, which was likely ascribed to the melt-recrystallization process of the α-phase into the SC phase, as found for the bulk polymer. The crystallinity (X_c_) degrees obtained for NP-HC-PLA and NP-SC-PLA were around 23 and 31%, respectively. Furthermore, the isothermal treatment at 65 °C of the doxorubicin-loaded nanoparticles revealed by DSC the crystallization of both NP-HC-PLA/DOX and NP-SC-PLA/DOX (Appendix A). However, a lower crystallization enthalpy was found with respect to the corresponding NP-HC-PLA and NP-SC-PLA counterparts (not drug-loaded), which is probably due to a partial crystallization of the NP upon formulation as a consequence of the nucleating effect of the doxorubicin, which is in agreement with WAXS analysis.

The nanoparticle nanostructures were subsequently monitored by WAXS measurements upon heating. NP-Rac-PLA retained the amorphous phase upon heating in agreement with the previous DSC analysis (Figure 5A). However, both NP-HC-PLA and NP-SC-PLA crystallized at 80 °C upon heating as evidenced by the development of WAXS reflections (Figure 5B,C) into the α-phase for the NP-HC-PLA ((110)/(200) and (203) (q (nm^−1^) = 11.9 and 13.6)) and the SC-phase for the NP-SC-PLA ((110), (300)/(030) and (220) (q (nm^−1^) = 8.5, 14.7 and 17)), indicating that the double-emulsion method did not impede the nanoparticles by possible degradation to further recrystallize upon heating. 

However, the annealing step to crystallize the freeze-dried nanoparticles leads to aggregation, deleting the pharmaceutical advantages of the nanoparticles for drug delivery applications. A simple method to prevent the aggregation of the NP-PLA upon heating is to disperse them in a matrix, which should enhance their hydrophilicity and ideally promote cellular growth as well as endorsing the DDS with storage capability and easy applicability. Gelatin is a natural polymer featuring inherent biocompatibility, uniformly distributed interconnected porosity and appreciable mechanical strength that is ideal for drug delivery and tissue engineering applications. A bionanocomposite hydrogel formed by NP-HC-PLA and fish gelatin was formed upon cooling to room temperature of the homogenized gelatin solution by sonication at 65 °C as the gelatin hydrogel network is disrupted, allowing the NP-PLA dispersion and the concomitant cold crystallization of the PLA nanoparticles in the α-phase, as confirmed by WAXS measurements at room temperature. The development of the α-crystalline phase reflections (Figure 6A) was observed over the WAXS profile of the gelatin, which features a broad amorphous halo with the superimposed signals of the triple helix with the characteristic domain distances of 11–12.6 Å and mainly the single α-helices with repeat distance of 2.9 Å (Appendix A). Furthermore, the NP-HC-PLA embedded in the gelatin hydrogels were isothermally annealed at 65 °C for 30 min, and the nanostructure development was probed by simultaneous SAXS/WAXS experiments (Figure 6B,D), confirming that the NP crystallinity could be customized. The crystallinity of the homo phase ((110)/(200) and (203) (q (nm^−1^) = 11.9 and 13.6), respectively) further increased during the isothermal treatment (Figure 6B), whilst the α-helices of the gelatin diminishes (Appendix A). Furthermore, the SAXS profile of the annealed bionanocomposite was exclusively sensible in the q range under study to the gelatin hydrogel nanostructure (Appendix A) due to the large size of the NP-PLA. Likewise, the SAXS intensity is dominated by hydrogel nanostructure as a result of the likely low scattering of NP-PLA internal structure considering its low concentration (Figure 6C,D). The measured SAXS profiles were fitted to fractal aggregates with a radius of gyration of ca. 1 nm and a fractal dimension of 1 with a correlation length of ca. 6 nm. In addition, the fibrillar structure of the triple helix undergoes a transition upon thermal treatment to a higher compact fractal cluster with a radius of gyration of firstly ca. 0.33 nm and then 0.65 with a fractal dimension of 2.5 and 3 and a correlation length of ca. 1 nm, indicating the transition of the gelatin chains to a random coil during the isothermal step at 65 °C (Appendix A). 

In addition, the effect of the nanoparticles embedded in the hydrogel thermal stability was assessed by DSC analyses. The sol–gel transition of the gelatin hydrogel network was shifted upon the incorporation of the NP-PLA from ca. 20 °C for the shark gelatin hydrogel to ca. 45–50 °C (Appendix A), which confirms that NP-PLA percolation throughout the hydrogel network reinforced the physical crosslinks, suggesting storage capability. 

The extended rheological properties of DOX-loaded bionanocomposites (GE/NP-PLA/DOX) were examined to assess in detail their mechanical performance by evaluating their critical deformation and storage capability. The stability of the bionanocomposites in the linear viscoelastic region (LVR) was probed by subjecting the GE/NP-PLA/DOX to strain sweeps at 20 °C, in which the strain amplitude ranged from 0.1 to 1000% at a constant temperature and a frequency of 10 rad·s^−1^ (Figure 7A). The strain sweeps were conducted to determine the region where the viscoelastic behavior of the gelatin network is independent of the applied strain. Likewise, the strain sweeps of the GE/NP-PLA/DOX have shown that the viscoelasticity remains fairly linear up to a strain greater than 100% (Figure 7A) and with higher values for the GE/NP-PLA/DOX compared to the hydrogel of gelatin, which proves the enhanced mechanical stability of the GE/NP-PLA/DOX. In addition, frequency sweeps of the GE/NP-PLA/DOX were performed to understand the viscoelastic behavior as a function of the relaxation time scale to allow the prediction of the storage properties and process conditions during the application of the materials (Figure 7B). The elastic modulus component prevailed over the viscous in the entire region (G′ > G″) under study [27], confirming its mechanical stability. The flow behavior of GE/NP-PLA/DOX indicates the formation of an increase in the elastic network, which is characteristic of complex gels that approach a solid-like trend. Both moduli increased with angular frequency, showing similar storage and loss modulus values between the studied gelatin networks, with GE/NP-Rac-PLA/DOX being the highest, followed by GE/NP-SC-PLA/DOX, indicating the greater stability of the gel.

The morphology of the gelatin hydrogel as well as the corresponding bionanocomposites was analyzed by cryo-SEM to probe the effect of the annealing step on both the NP-PLA and the gelatin hydrogel network (Appendix A). A porous network was found for both the shark gelatin hydrogel and the bionanocomposites in agreement with previous tuned gelatin hydrogel [27]. However, the size of the pores showed that the NP-PLA promoted the development of a mesh with bimodal cavity distribution in which a second network features a finer pore dimension (Appendix A), whereas the gelatin hydrogel network (GE) presents a standard lognormal size distribution. Furthermore, the lack of NP-PLA clusters within the gelatin hydrogel network proved the efficacy to avoid the NP agglomeration upon annealing by the gelatin matrix.

In addition, the so-called “syringe test” [43] was conducted to evaluate the mechanical behavior of hydrogels during a local administration through a needle. The rheological response in terms of the apparent viscosity of the bionanocomposites was assessed with three different steps (two different shear rates and two different temperatures; see the Experimental section). The shark gelatin hydrogel (GE) was characterized by an initial viscosity of 10.1 ± 2.1 Pa·s (*p* < 0.03) at a shear rate of 0.1 s^−1^ (Figure 8A, blue box), whilst the addition of the nanoparticles into the GE increased the viscosity of the bionanocomposite to 219.6 ± 11.0 Pa·s, 488.6 ± 62.9 Pa·s and 1720 ± 115.58 Pa·s (*p* < 0.03) for GE/NP-Rac-PLA/DOX, GE/ NP-SC-PLA/DOX and GE/ NP-HC-PLA/DOX, respectively (Figure 8B–D, blue box), indicating that NP-PLA reinforced the three-dimensional network of the gelatin hydrogel and in agreement with the increase in the sol/gel transition upon the addition of the nanoparticles found by DSC. Subsequently, the shear rate was increased to 100 s^−1^, and the temperature was gradually raised to 37 °C (Figure 8A–D, green box), simulating an injection through a needle of 0.9 mm in diameter and the temperature gradient (from 20 to 37 °C) along the needle in contact with the human body. The viscosity of all hydrogels decreased below 5 Pa·s when the shear rate was set to 100 s^−1^, 1.9 ± 1.1 Pa·s, 1.25 ± 0.4 Pa·s, 1.9 ± 1.2 and 2.3 ± 1.1 Pa·s for GE, GE/NP-Rac-PLA/DOX, GE/NP-SC-PLA/DOX and GE/NP-HC-PLA/DOX, respectively, and it gradually further decreased upon the increase in temperature, indicating that all the gelatin-based hydrogels can be easily injected despite the initially high viscosity. Finally, the shear rate was set at 0.1 s^−1^ at a constant temperature of 37 °C, simulating physiological conditions (Figure 8A–D, purple box). The viscosity of the GE hydrogel slightly increased with time, whilst the GE hydrogels containing NP-PLA exhibited an immediate increase in viscosity upon reducing the shear rate followed by an increase upon time. The viscosity values of the bionanocomposites upon injection are close to the viscosity values of the human liver, 1.9 ± 0.9 Pa·s, 21.3 ± 3.8 Pa·s, 4.7 ± 1.2 and 5.7 ± 1.6 Pa·s for GE, GE/NP-Rac-PLA/DOX, GE/NP-SC-PLA/DOX and GE/NP-HC-PLA/DOX, respectively, turning the GE hydrogel containing NP into a potential biomaterial for liver tissue regeneration [59] as well as for the distribution into a complex anatomical human body area in particular after tumor resection where a combination of therapeutic agents is required to be administrated with low invasive techniques and that concomitantly promote the cellular growth. 

Finally, the release of the loaded doxorubicin from the NP-PLA/DOX as well as from the GE/NP-PLA/DOX was monitored in PBS at 37 °C by UV-Vis spectrometry (Figure 9). Shark gelatin absorption spectra present a UV-Vis band at 215 nm with a shoulder at 290 nm, whilst the doxorubicin and PLA main absorption bands are located at 490 nm and 231 nm, respectively [60], which permits the monitoring of the drug release profiles. All the NP-PLA/DOX (NP-HC, NP-SC and NP-rac) released completely the loaded doxorubicin within a few hours, with a significant burst release step during the first 2 h (98.9%, 60.9% and 51.6% for NP-HC, NP-Rac and NP-SC, respectively, Figure 9) that might be related to the formation of internal NPs microvoids due to heterogeneous structural relaxation during physical aging. The microvoids could be then filled with water when exposed to an aqueous solution that promotes the doxorubicin dissolution from the inner NP region [61]. Moreover, the enthalpy relaxation values related to physical aging found by DSC (1.44 J/g, 1.2 J/g and 1.06 J/g for NP-HC, NP-Rac and NP-SC, respectively, Appendix A) are in agreement with the burst release differences between formulations, being both the enthalpy and the burst release higher for NP-HC followed by NP-Rac (Figure 10). 

In contrast, the doxorubicin release from the GE/NP-PLA/DOX exhibited a sustained pattern similar to all the NP formulations (≈5% in 120 h) that was slower than the doxorubicin release from the gelatin network itself, confirming the tailoring of the drug release profiles. Moreover, GE/NP-SC-PLA/DOX exhibited slightly higher release, followed by GE/NP-HC-PLA/DOX and GE/NP-Rac-PLA/DOX, which might be related to the crystallinity difference found by DSC (Appendix A), since the doxorubicin molecule might interact preferentially with the amorphous phase of the nanoparticles, resulting in a higher doxorubicin/amorphous phase ratio for the more crystalline NP, namely GE/NP-SC-PLA/DOX followed by GE/NP-HC-PLA/DOX, and hence, a slightly higher release rate (Figure 10).

## 4. Conclusions

A straightforward methodology to anneal the PLA nanoparticles to customize the drug release rate was developed by generating bionanocomposites with shark gelatin hydrogels. The nanoparticles were successfully formulated with different PLA derivatives and loaded with two different antitumoral compounds but yielded amorphous PLA materials upon formulation, erasing the designed nanostructures and their different release rate. The dispersion of PLA nanoparticles in the gelatin hydrogel permits their thermal annealing and thus, their crystallization, without the detrimental agglomeration that diminishes their therapeutic effect. The bionanocomposites generated were fully characterized by a multi-technique approach to reveal the nanostructure of the formulated nanoparticles as well as their development during the thermal annealing within the gelatin hydrogel network. The SAXS and WAXS profiles as well as the cryo-SEM, DSC and rheological measurements of the bionanocomposites suggested the stability and reversibility of the hydrogel network upon the incorporation of the nanoparticles. The bionanocomposites generated were proved to feature storage capability at room temperature as well as being able to be injected by a syringe (tested by the so-called “syringe test”), which provides the possibility to be administrated locally that minimizes the detrimental invasive effects.

The results suggest that the generated bionanocomposites based on gelatin and comprising PLA nanoparticles with several nanostructures might have potential in the development of DDS for local release in soft tissue treatments. In addition, the gelatin hydrogel was previously proved to be formulated with different model therapeutic agents such as doxorubicin and crocin, extending its use as merely support to co-vehicle complementary therapeutic agents and design complex treatments with different release rates. Furthermore, the gelatin enhances the biocompatibility of the PLA nanoparticles as well as promotes cellular growth, endorsing the drug delivery system with multifunctionality applications. Moreover, the one-step methodology to crystallize the NP simultaneously with the bionanocomposite formation would result in resource savings at the industrial level. Furthermore, the multiprofile drug release obtained from the different components of the bionanocomposite could be applied for the treatment of complex tumors that need the administration of alternative drugs, such as the urotherial carcinoma. Future work is required to investigate the influence of the NP concentration in the gelatin matrix as well as the effect of the bionanocomposite on cell viability by *in vitro* models.

## Figures and Tables

**Figure 1 pharmaceutics-14-01138-f001:**
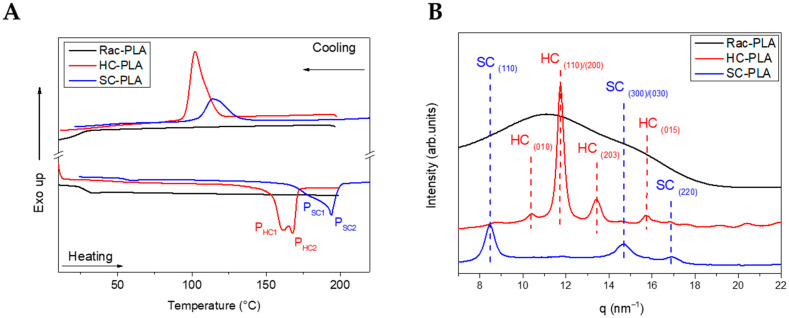
DSC thermograms (**A**) and WAXS patterns (**B**) of bulk polymers at room temperature.

**Figure 2 pharmaceutics-14-01138-f002:**
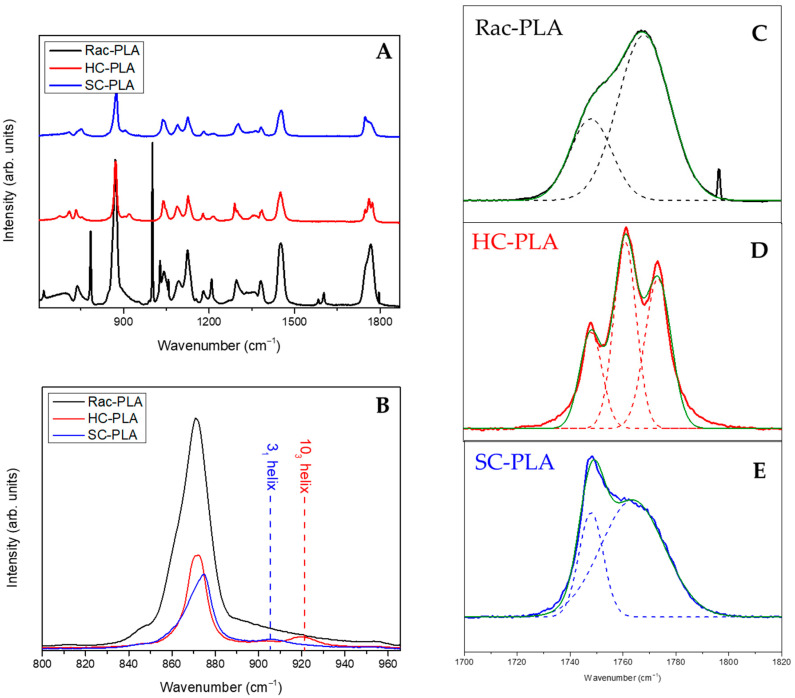
Full Raman spectrum of the bulk polymers (**A**), C-COO stretching Raman region of the bulk polymers (**B**) and carbonyl region of the Rac-PLA (**C**), HC-PLA (**D**) and SC-PLA (**E**).

**Figure 3 pharmaceutics-14-01138-f003:**
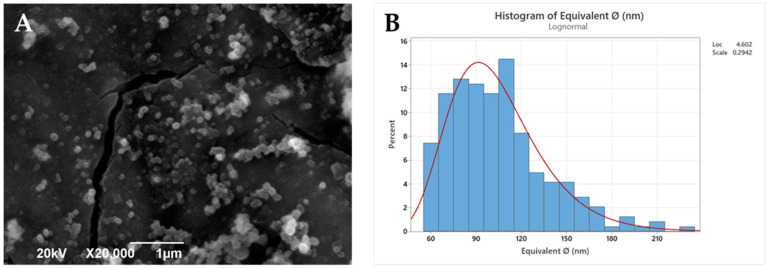
SEM image (**A**) and histogram (**B**) of the statistical diameter analysis.

**Figure 4 pharmaceutics-14-01138-f004:**
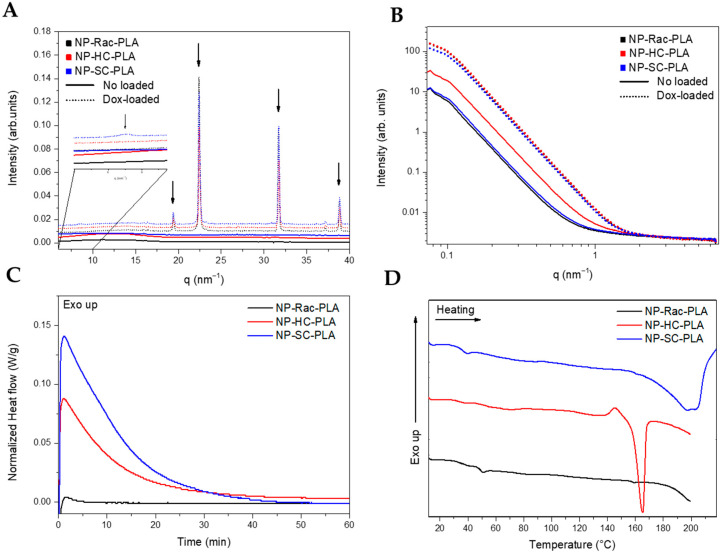
WAXS (**A**) and SAXS (**B**) of the different PLA nanoparticles. DSC thermograms of the nanoparticles during the isothermal step at 65 °C (**C**) and subsequent heating (**D**).

**Figure 5 pharmaceutics-14-01138-f005:**
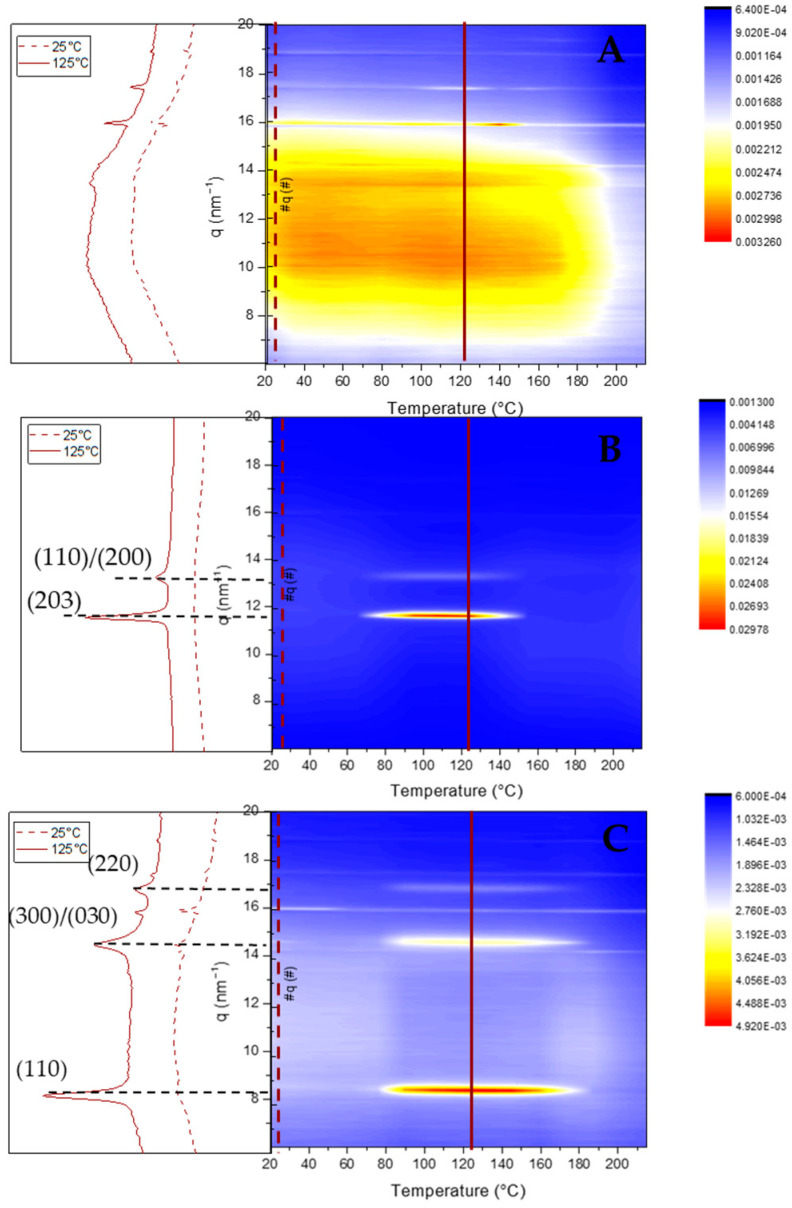
WAXS patterns of NP-Rac-PLA (**A**), NP-HC-PLA (**B**) and NP-SC-PLA (**C**) upon heating.

**Figure 6 pharmaceutics-14-01138-f006:**
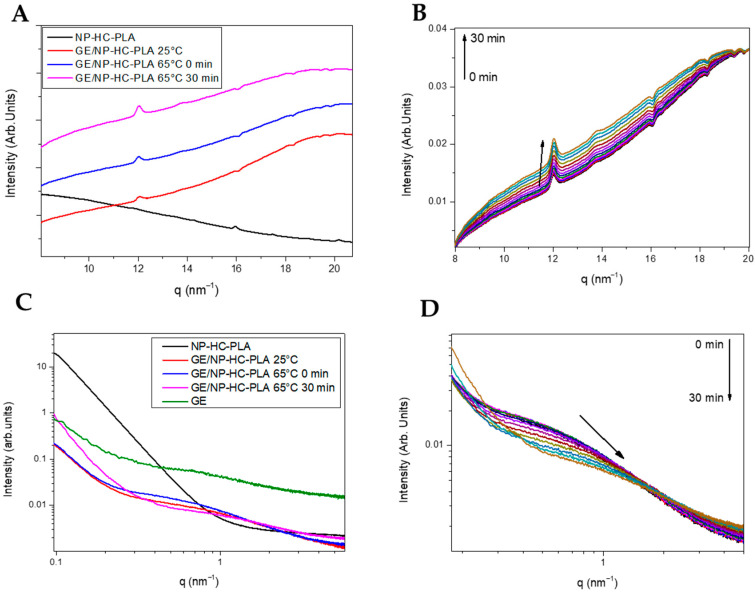
(**A**) WAXS pattern of NP-HC-PLA (black line), the bionanocomposite of NP-HC-PLA at room temperature (red line), 65 °C (blue line) and after 30 minutes at 65 °C (pink line). (**B**) WAXS pattern evolution during the isothermal at 65 °C. (**C**) SAXS pattern of NP-HC-PLA (black line), the bionanocomposite of NP-HC-PLA at room temperature (red line), 65 °C (blue line) and after 30 minutes at 65 °C (pink line). (**D**) SAXS pattern evolution during the isothermal at 65 °C.

**Figure 7 pharmaceutics-14-01138-f007:**
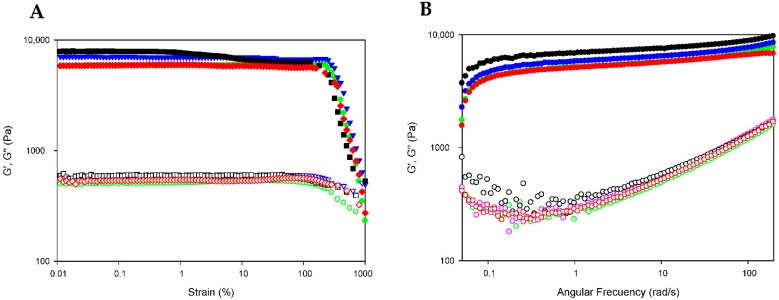
Strain and frequency sweep for GE (green color), GE/NP-SC-PLA/DOX (blue color), GE/NP-HC-PLA/DOX (red color) and GE/NP-Rac-PLA/DOX (black color). Storage (G′; filled symbols) and loss moduli (G″; hollow symbols) depicted versus strain (**A**) and angular frequency (**B**) at 20 °C.

**Figure 8 pharmaceutics-14-01138-f008:**
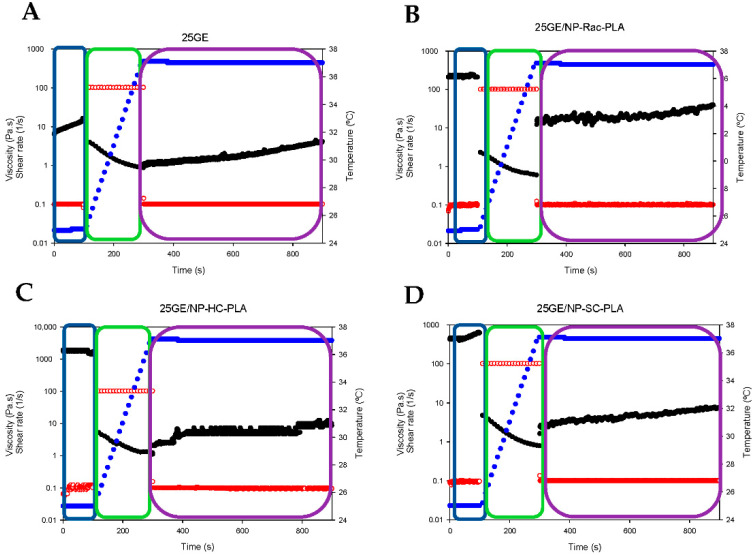
Syringe test results of gelatin hydrogel (**A**), GE/NP-Rac-PLA/DOX (**B**), GE/NP-HC-PLA/DOX (**C**) and GE/NP-SC-PLA/DOX (**D**). Viscosity (black color), shear rate (red color), temperature (blue color).

**Figure 9 pharmaceutics-14-01138-f009:**
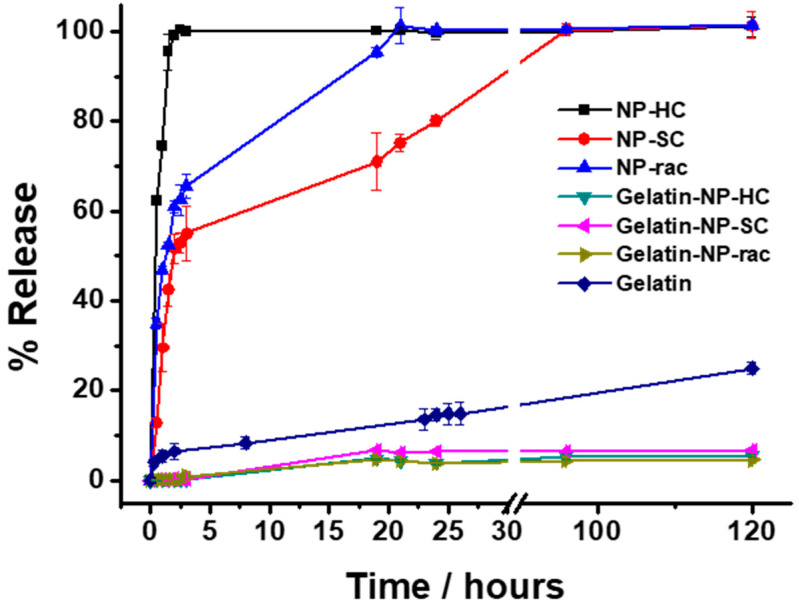
Release profiles of DOX from nanoparticles in suspension (NP-HC, NP-SC and NP-rac), gelatin (Gelatin) and nanoparticles embedded in gelatin (Gelatin-NP-HC, Gelatin-NP-SC and Gelatin-NP-Rac) in PBS pH 7.4 at 37 °C. The drug releases were tested in three replicates. Error bars are 2σ.

**Figure 10 pharmaceutics-14-01138-f010:**
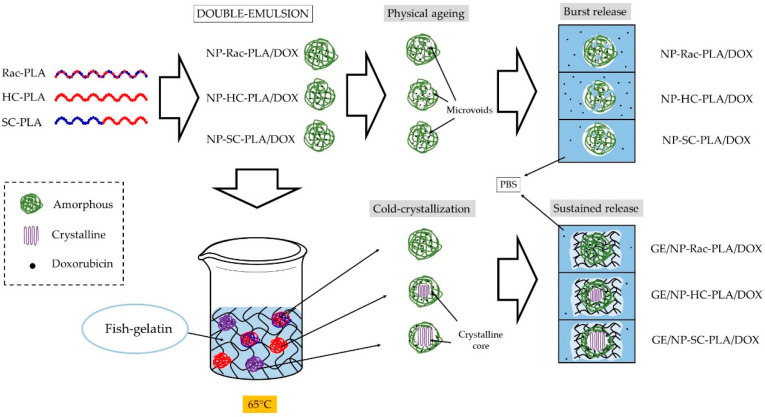
Mechanism of bionanocomposites formation and the drug release.

**Table 1 pharmaceutics-14-01138-t001:** Bulk polymer characterization.

Sample	Name	M_n_ (g/mol) ^A^	PDI ^B^	T_g_ (°C) ^C^	T_c_ (°C) ^D^	ΔH_c_ (J/g) ^E^	T_m_ (°C) ^F^	ΔH_m_(J/g) ^G^	X_c_(%) ^H^	P_m_ ^I^	[α] ^J^
PDLLA	Rac-PLA	59.594	1.77	24.12	-	-	-	-	-	0.52	−1.1
PLLA	HC-PLA	36.047	1.92	40.84	101.24	46.102	170.24	49.538	53.75	0.99	−159.4
PLLA-*b*-PDLA	SC-PLA	33.238	2.38	41.28	113.83	44.767	193.93	46.91	33.03	0.90	−10.7

^A^ Molecular weight in number average obtained by GPC relative to polystyrene standards in chloroform. ^B^ Polydispersity Index obtained by GPC. ^C^ Glass transition temperature. ^D^ Crystallization temperature. ^E^ Crystallization enthalpy. ^F^ Melting temperature. ^G^ Melting enthalpy. ^H^ Crystallinity $X_{c}=\frac{\Delta H_{m}}{\Delta H_{f}^{0}}\times 100$ with $\Delta H_{mHC}^{0}=(93.7\;J/g$, crystallinity) and $\Delta H_{mSC}^{0}=(142\;J/g$, crystallinity). ^I^ Probability of finding meso tetrads calculated from homonuclear decoupling ^1^H NMR spectra after deconvolution; calculations based on CEC statistics [45]. ^J^ Specific optical rotation ((α)_PLLA_= −173°).

**Table 2 pharmaceutics-14-01138-t002:** DLS characterization of PLA nanoparticles.

Sample	Average Diameter (nm)	PDI	Z Potential (mV)
NP-Rac-PLA	231.52 ± 11.18	0.11 ± 0.02	−14.99 ± 2.02
NP-HC-PLA	239.64 ± 19.16	0.10 ± 0.03	−14.56 ± 3.62
NP-SC-PLA	256.8 ± 9.19	0.09 ± 0.03	−10.00 ± 2.60

## Data Availability

Not applicable.

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
