# Peer review of "Multifunctional PLA/Gelatin Bionanocomposites for Tailored Drug Delivery Systems"

_pharmaceutics, 2022, doi:10.3390/pharmaceutics14061138_

Round 1

Reviewer 1 Report

General comments

The manuscript “Multifunctional PLA/Gelatin bionanocomposites for tailored 2 drug delivery systems” is quite interesting and good for scientific communities including drug delivery. However I reviewed and found some major suggestions for improvements of the quality of manuscript.

These are the some major suggestions:

  1. Authors should go whole manuscript and correct the Grammar and spelling mistake e.g...hydrophicity, fisiological etc in introduction section.
  2. They have expand on first the abbreviation where used.
  3. In preparation of PLA-nanoparticles, authors should provide details information of the manufacturing process like evaporation time, temperature of water bath.
  4. On line 180, authors should mentioned or explained about 25GE.
  5. On line 182, authors should mentioned the ratio of 0.4/25/74.6 is taken either in w/w/ or v/v or in percentage.
  6. “DLS measurement” should be replaced by “DLS technique”.
  7. On line 317, authors should explain about Pm value on first time.
  8. In DSC chromatogram, authors should revise and rewrite the results as they wrongly discuss the exothermic and endothermic characteristics, so this thermogram need to revise thoroughly
  9. In fig 2 description of Fig 2B is missing.
  10. Authors should add the well justification related to particle size decreasing upon lyophilization. Shrinking effects is not proper justification.
  11. Fig 4. The DSC thermogram of the drug loaded nanoparticles is missing. Also Fig 4 D the TG of the NP-Rac-PLA is not visible. And the double is also not clearly visible in NP-SC-PLA.

Author Response

            REVIEWER 3

The manuscript “Multifunctional PLA/Gelatin bionanocomposites for tailored 2 drug delivery systems” is quite interesting and good for scientific communities including drug delivery. However I reviewed and found some major suggestions for improvements of the quality of manuscript.

These are the some major suggestions:

  1. Authors should go whole manuscript and correct the Grammar and spelling mistake e.g...hydrophicity, fisiological etc in introduction section.

The authors thank the comment of the reviewer and have checked the grammar and spelling mistakes throughtout the manuscript that have been corrected.

  1. They have expand on first the abbreviation where used.

The authors thank the reviewer comment and have added the abbreviations when missing.

  1. In preparation of PLA-nanoparticles, authors should provide details information of the manufacturing process like evaporation time, temperature of water bath.

The authors thank the reviewer comment and we agree about the importance to include the details about the PLA-nanoparticles manufacturing method. The experimental procedure was added in the materials and methods section as follows:

“Nanoparticles formulation. PLA-Nanoparticles (NP-PLA) were formulated from the three PLA derivatives, racemic-PLA (NP-rac-PLA), homopolymer-PLA (NP-HC-PLA) and stereo-diblock-PLA (NP-SC-PLA). In addition, all the PLA-derivatives NP were loaded with the antitumoral doxorubicin and dasatinib as reference drugs [34–36]. Nanoparticles were prepared following the double emulsion methodology. Briefly, 10 mg of PLA were dissolved in 4 mL of CH2Cl2 and 1 mL of mQ water was added, the mixture was shaken in a Vortex and subsequently sonicated in an ice bath for 1 min, to avoid emulsion overheating. The pre-emulsion generated was added to 10 mL of PVA (1%), shaken in a Vortex and sonicated for 5 min. The emulsion was allowed to evaporate organic solvent at room temperature for 45 min in a fume hood, centrifuged at 4°C and 15k rpm for 20 minutes. Nanoparticles were collected and dissolved in PBS pH=7. Drug-encapsulated nanoparticles were prepared following the methodology above-mentioned. The amount of the drug (1 mg) was dissolved along with the polymer in 4 mL of CH2Cl2. The encapsulated drug in the NP-PLA is indicated at the end of the abbreviation name: NP-rac-PLA/DOX. The correspondence between the sample abbreviations and their compositions is summarized in the Supplementary Information (Table S1).”

In addition, the performance of the designed nanoparticles has been related to the nanostructure obtained throughout the whole process.

  1. On line 180, authors should mentioned or explained about 25GE.

The authors thank the suggestion made by the reviewer and have changed “25GE” to only “GE” for the shake of clearness. The 25 preceding the GE refers to the proportion of Gelatin in the bionanocomposite composition, however, it has been fixed for all the bionanocomposites so it does not add any value to the herein manuscript.

  1. On line 182, authors should mentioned the ratio of 0.4/25/74.6 is taken either in w/w/ or v/v or in percentage.

The authors thank the comment of the reviewer and we have detailed the composition ratio of the bionanocomposite as follow (w/w) in the materials and method section.

  1. “DLS measurement” should be replaced by “DLS technique”.

The authors thank to spot the wrong terminology and we have changed the suggested title in the materials and method section as DLS technique

  1. On line 317, authors should explain about Pm value on first time.

The authors thank the reviewer to have spotted the lack of clearness about the terminology. We have modified the text for the shake of clearness as follows:

Furthermore, the Pm values, known as the probability of isotactic enchainment being Pm =0.5 a random insertion whilst isotactic enchainment is designated by Pm=1 [49], were obtained from deconvoluted homonuclear decoupling 1H NMR spectra (Figure S4-S6) and confirmed the atactic character of the rac-PLA and the isotactic character of HC-PLA and SC-PLA (Table 1).

We have included a reference for further description.

  1. In DSC chromatogram, authors should revise and rewrite the results as they wrongly discuss the exothermic and endothermic characteristics, so this thermogram need to revise thoroughly

The authors thank the reviewer to have spotted the wrong uses of the endothermic and exothermic terms and we have changed them throughout the manuscript.

  1. In fig 2 description of Fig 2B is missing.

The authors thank the reviewer to have realised the lack of proper legend for Figure 2. We have added the missing information to the Figure 2 description.

  1. Authors should add the well justification related to particle size decreasing upon lyophilization. Shrinking effects is not proper justification.

The authors thank the comment of the reviewer and have included in the explanation the likely swelling of nanoparticles in solution which is measured by DLS analysis.

  1. Fig 4. The DSC thermogram of the drug loaded nanoparticles is missing. Also Fig 4 D the TG of the NP-Rac-PLA is not visible. And the double is also not clearly visible in NP-SC-PLA.

The authors agree with the reviewer that the Tg and the double melting peak of the NP-Rac-PLA and NP-SC-PLA, respectively, are not clearly visible and have added the zoom-in regions in the supplementary information for the shake of clarity. In addition, we have performed new DSC measurements for the drug-loaded nanoparticles and included them in the manuscript.

Reviewer 2 Report

The topic submitted is novel and adds significant research data to the existing field of research. The article is not very well articulated needs English language revisions and even formatting of the manuscript as per the MDPI guidelines. The manuscript needs to be checked for statistical significance Fig 7. The introduction needs to be concise. Abstract and conclusion should include a sentence proposing the future direction of the present research. Also the commercial aspects of how industries can benefit.  Also, the composite application in a drug delivery system using an in-vitro model to understand drug release is advisable. The conclusion is very lenghty needs to be concise.  Most of the methodology lacks citing the references from where the technique was adopted and used to conduct the study. Below are a few relevant article suggestions that authors are recommended to cite in the methodology, results /discussion section to justify and strengthen the research data obtained.  

Food Research International 51, no. 2 (2013): 714-722.

Gels 8, no. 2 (2022): 103.

 Antioxidants10(12), p.1976.

Pharmaceutics 13, no. 6 (2021): 848.

Journal of Polymer Research25(10), pp.1-12.

Author Response

REVIEWER 1

  1. The topic submitted is novel and adds significant research data to the existing field of research. The article is not very well articulated needs English language revisions and even formatting of the manuscript as per the MDPI guidelines.

The authors agree about the English level improvement of the manuscript. The text of the manuscript has been amended to correct language mistakes as well as to improve the narrative flow to be clearer. Moreover, the manuscript has been formatted to better adjust to the MDPI guidelines, for example the reference list has been adjusted to the MDPI style.

  1. The manuscript needs to be checked for statistical significance Fig 7.

We thank the observation of the reviewer about the statistical correctness dismissal and we have included it throughout the manuscript to improve the quality of the experimental work presented in the manuscript. We have also added in the materials and method section the details regarding the experimental repeatability of several techniques.

Changes in the manuscript:

Lines: 284-285

Experiments were repeated three times to analyze the reproducibility of obtained results and the deviations in (complex) viscosities between repetitions were lower than 3%.

Viscosity measurement deviations were added to the text:

Lines: 534-538

The shark gelatin hydrogel (GE) was characterized by an initial viscosity of 10.1 ± 2.1 Pa·s (p<0.03) at a shear rate of 0.1 s-1 (Figure 7 A-D, blue box), whilst the addition of the nanoparticles into the GE increased the viscosity of the bionanocomposite to 219.6 ± 11.0 Pa·s, 488.6 ± 62.9 Pa·s and 1720 ±115.58 Pa·s (p<0.03) for NP-Rac-PLA, NP-SC-PLA and NP-HC-PLA, respectively.

Lines:545-547

The viscosity of all hydrogels decreased below 5 Pa·s when the shear rate was set to 100 s−1, 1.9 ± 1.1 Pa·s, 1.25 ± 0.4 Pa·s, 1.9 ± 1.2 and 2.3 ± 1.1 Pa·s for GE, NP-Rac-PLA, NP-SC-PLA and NP-HC-PLA, respectively.

Lines:553-556

 The viscosity values of the bionanocomposites upon injection are close to the viscosity values of the human liver, 1.9 ± 0.9 Pa·s, 21.3 ± 3.8 Pa·s, 4.7 ± 1.2 and 5.7 ± 1.6 Pa·s for GE, NP-Rac-PLA, NP-SC-PLA and NP-HC-PLA, respectively.

  1. The introduction needs to be concise.

The authors thank the comment of the reviewer and have changed the structure of the Introduction to be concise by eliminating information that do not add relevant value to the manuscript, suchs as :

“Generally, semicrystalline polymers trapped in the kinetic state due to the quenching during processing conditions, are annealed to recover the crystalline domain through the cold-crystallization transition. However, the low Tm of PEG (50°C) comparable to the PLA Tg (50-65°C) compromise the integrity of the nanoparticles by their aggregation, for the common approach to enhance the PLA hydrophicity by copolymerized PLA with PEG”

However, some other information has been added to answer other reviewers comment. The authors have tried to be the more concise and clear possible while satisfying all the reviewers requests.

  1. Abstract and conclusion should include a sentence proposing the future direction of the present research. Also the commercial aspects of how industries can benefit. Also, the composite application in a drug delivery system using an in-vitro model to understand drug release is advisable.

The authors thank the reviewer comment and have included at the end on the conclusion the future direction of the research as well as the commercial aspects for the industrial benefit:  “Moreover, the one-step methodology to crystallize the NP simultaneously with  the bionanocomposite formation would result in resource savings at the industrial level. Furthermore, the multiprofile drug release obtained from the different components of the bionanocomposite could be apply for the treatment of complex tumours that need the administration of alternately drugs, such as for the urotherial carcinoma. Future work is required to investigate the influence of the NP concentration in the gelatin matrix as well as the effect of the bionanocoposite in the cell viability in in vitro models”

  1. The conclusion is very lenghty needs to be concise. 

The authors thank the comment of the reviewer and have summarized the conclusions highlighting also the main new scientific achieved goals with the extra performed experiments as follows:

“A straightforward methodology to anneal the PLA nanoparticles to customise the drug release rate was developed by generating bionanocomposites with shark gelatin hydrogels. The nanoparticles were successfully formulated with different PLA derivatives and loaded with two different antitumoral compounds but yielded amorphous PLA materials upon formulation, erasing the designed nanostructures and their different release rate. The dispersion of PLA nanoparticles in the gelatin hydrogel permits their thermal annealing and thus, their crystallization, without the detrimental agglomeration that diminishes their therapeutic effect. The bionanocomposites generated were fully characterized by a multi-technique approach to reveal the nanostructure of the formulated nanoparticles as well as their development during the thermal annealing within the gelatin hydrogel network. The SAXS and WAXS profiles as well as the DSC and rheological measurements of the bionanocomposites suggested the stability and reversibility of the hydrogel network upon the incorporation of the nanoparticles. Besides, the bionanocomposites generated were proved to feature storage capability at room temperature as well as being able to be injected by a syringe (tested by the so-called “syringe test”) which provides the possibility to be administrated locally that minimise the detrimental invasive effects.

The results suggest that the generated bionanocomposites based on gelatin and comprising PLA nanoparticles with several nanostructures might have potential in the development of DDS for local release in soft tissue treatments. In addition, the gelatin hydrogel was previously proved to be formulated with different model therapeutic agents such as doxorubicin and crocin, extending its use as merely support to co-vehicle complementary therapeutic agents and design complex treatments with different release rates. Furthermore, the gelatin enhance the biocompatibility of the PLA nanoparticles as well as promotes cellular growth endorsing the drug delivery system with multifunctionality applications. Moreover, the one-step methodology to crystallize the NP the same time as the bionanocomposite is formed would result in resource savings at the industrial level. Future work is required to investigate the influence of the NP concentration in the gelatin matrix as well as the effect of the bionanocoposite in the cell viability in in vitro models.”

  1. Most of the methodology lacks citing the references from where the technique was adopted and used to conduct the study. Below are a few relevant article suggestions that authors are recommended to cite in the methodology, results /discussion section to justify and strengthen the research data obtained.  

Food Research International 51, no. 2 (2013): 714-722.

Gels 8, no. 2 (2022): 103.

 Antioxidants10(12), p.1976.

Pharmaceutics 13, no. 6 (2021): 848.

Journal of Polymer Research25(10), pp.1-12.

The authors thank the comment of the reviewer and have included in the manuscript most of the suggested references.

Reviewer 3 Report

This paper is the investigation of gelatin/PLA nanocomposite particles for the drug delivery system. This is interesting and valuable for researchers on DDS, biomaterials, or pharmaceutics. However, some descriptions are lacking. For example, the properties or characteristics of gelatin and PLA should be introduced in detail. What is the advantage compared to other biomaterials, such as alginate, collagen, PLGA, and so on? The authors should introduce the gelatin-based drug release should be introduced. In addition, how about the ratio of gelatin/PLA? How about the release kinetics of DOX from the nanocomposite? What is the strength or novelty of the nanocomposite compared to other biomaterials?

The presentation quality is poor, and it is difficult for readers to understand the manuscript. Taken together, many major revisions are recommended to expect the authors’ significant improvement. The paper would be re-considered only when all the comments were responded.

  1. Lines 96-101 or Discussion

The information of gelatin, such as drug release carrier or application example, should be introduced. In the current manuscript, the readers cannot understand the gelatin properties.

To reduce the authors’ burden, I suggest the sentences or references to be added for the revision.

Information

doi.org/10.1016/j.foodhyd.2011.02.007

Characterization as drug carrier

doi.org/10.1016/S0169-409X(97)00125-7

Molecules 202126(22), 6795

Representative application (review and research)

a. wound healing

doi.org/10.1016/j.ijbiomac.2019.07.155

doi.org/10.1016/j.bone.2008.06.019

b. cancer therapy

doi.org/10.1016/j.jconrel.2013.09.019

doi.org/10.2217/nnm-2018-0028

c. 3D cell culture

Cancers 202012(10), 2754

Tissue Eng. Part C Methods 201925, 711–720. https://doi.org/10.1089/ten.tec.2019.0189

  1. Lines 101-108

What is the strength and novelty of the PLA/gelatin-based scaffold? It is difficult to understand the material. Therefore, the authors should introduce or discuss the biomaterial-based scaffold and compare it with them. To reduce the authors’ burden, I suggest the sentences or references to be added for the revision.

Overall

Int. J. Mol. Sci. 202122(16), 8657

Alginate

Materials 20136(4), 1285-1309

Collagen

Polymers 20168(2), 42

PLGA

doi.org/10.1016/j.msec.2015.11.026

  1. Materials and Methods

The new section should be indicated clearly in a bold type.

  1.  

The authors should indicate the DOX release kinetics.

  1. Table 2

The figures should express the mean ± the standard error.

  1. Figure 3

The authors should show the inner structure.

7.

There are many hard-to-sentences. The written English of the manuscript should be considerably improved, and therefore, a revision by a native speaker is highly recommended.

  1. Is the PLA/gelatin ratio affected the results? The authors should discuss the points.

9.

Why is the DOX selected? Is it possible to apply cancer therapy? Is the result different among the drugs?

10.

How about the degradation profiles of the nanocomposite particles?

  1.  

Considering the inflammation induction or property control, PLGA must be appropriate material than PLA. Why is the PLA selected? Overall, the scaffold strength or novelty is not clear.

Author Response

REVIEWER 2

This paper is the investigation of gelatin/PLA nanocomposite particles for the drug delivery system. This is interesting and valuable for researchers on DDS, biomaterials, or pharmaceutics. However, some descriptions are lacking. For example, the properties or characteristics of gelatin and PLA should be introduced in detail. What is the advantage compared to other biomaterials, such as alginate, collagen, PLGA, and so on? The authors should introduce the gelatin-based drug release should be introduced. In addition, how about the ratio of gelatin/PLA? How about the release kinetics of DOX from the nanocomposite? What is the strength or novelty of the nanocomposite compared to other biomaterials?

The presentation quality is poor, and it is difficult for readers to understand the manuscript. Taken together, many major revisions are recommended to expect the authors’ significant improvement. The paper would be re-considered only when all the comments were responded.

  1. Lines 96-101 or Discussion. The information of gelatin, such as drug release carrier or application example, should be introduced. In the current manuscript, the readers cannot understand the gelatin properties.

To reduce the authors’ burden, I suggest the sentences or references to be added for the revision.

Information

doi.org/10.1016/j.foodhyd.2011.02.007

Characterization as drug carrier

doi.org/10.1016/S0169-409X(97)00125-7

Molecules 202126(22), 6795

Representative application (review and research)

  1. wound healing

doi.org/10.1016/j.ijbiomac.2019.07.155

doi.org/10.1016/j.bone.2008.06.019

  1. cancer therapy

doi.org/10.1016/j.jconrel.2013.09.019

doi.org/10.2217/nnm-2018-0028

  1. 3D cell culture

Cancers 202012(10), 2754

Tissue Eng. Part C Methods 201925, 711–720. https://doi.org/10.1089/ten.tec.2019.0189

 The authors thank the comment of the reviewer as well as the suggested literature to improve the manuscript. The text of the manuscript has been amended to correct language mistakes as well as to improve the narrative flow to be clearer The authors have extended in particular, the gelatin general information in the main text of the manuscript as well as performed extra rheological measurements to characterise the gel network strength and cryo-SEM experiments to probe the morphological aspects of the hydrogel network.

  1. Lines 101-108. What is the strength and novelty of the PLA/gelatin-based scaffold? It is difficult to understand the material. Therefore, the authors should introduce or discuss the biomaterial-based scaffold and compare it with them. To reduce the authors’ burden, I suggest the sentences or references to be added for the revision.

 Overall

Int. J. Mol. Sci. 202122(16), 8657

Alginate

Materials 20136(4), 1285-1309

 Collagen

Polymers 20168(2), 42

 PLGA

doi.org/10.1016/j.msec.2015.11.026

The authors thank the reviewer comments and specific recommendations, and have added the requested information in the introduction:

“Several bio-based materials have been previously used for the fabrication of scaffolds such as collagen, chitosan, alginate, gelatin, silk, etc.[18] Alginate is an attractive material for biomedical applications due to the structural similarities to polysaccharides, but it is still unable to meet the required parameters for tissue engineering (degradation, mechanical properties, bioactivity) [19]. Collagen-based scaffolds for the regeneration of tissues such as nerve, bone, cartilage or skin are also been developed, however, the low solubility in water turns gelatin into a better candidate for tissue engineering [20]”.

  1. Materials and Methods. The new section should be indicated clearly in a bold type.

The authors thank the comment of the reviewer and have modified the material and method section accordingly.

  1. The authors should indicate the DOX release kinetics.

The authors agree about the need for the release of kinetics experiments to enhance the scientific outcome of the manuscript. The release kinetics of doxorubicin from the PLA NP as well as within the gelatin hydrogel network and finally from the PLANP within the hydrogel network has been monitored. The results have been compared to strengthen the applications of the designed DDS system.

  1. Table 2. The figures should express the mean ± the standard error.

The authors thank the reviewer comment and have added the standard errors to the manuscript as follows:

Lines: 284-285

Experiments were repeated three times to analyze the reproducibility of obtained results and the deviations in (complex) viscosities between repetitions were lower than 3%.

Viscosity measurement deviations were added to the text:

Lines: 534-538

The shark gelatin hydrogel (GE) was characterized by an initial viscosity of 10.1 ± 2.1 Pa·s (p<0.03) at a shear rate of 0.1 s-1 (Figure 7 A-D, blue box), whilst the addition of the nanoparticles into the GE increased the viscosity of the bionanocomposite to 219.6 ± 11.0 Pa·s, 488.6 ± 62.9 Pa·s and 1720 ±115.58 Pa·s (p<0.03) for NP-Rac-PLA, NP-SC-PLA and NP-HC-PLA, respectively.

Lines:545-547

The viscosity of all hydrogels decreased below 5 Pa·s when the shear rate was set to 100 s−1, 1.9 ± 1.1 Pa·s, 1.25 ± 0.4 Pa·s, 1.9 ± 1.2 and 2.3 ± 1.1 Pa·s for GE, NP-Rac-PLA, NP-SC-PLA and NP-HC-PLA, respectively.

Lines:553-556

 The viscosity values of the bionanocomposites upon injection are close to the viscosity values of the human liver, 1.9 ± 0.9 Pa·s, 21.3 ± 3.8 Pa·s, 4.7 ± 1.2 and 5.7 ± 1.6 Pa·s for GE, NP-Rac-PLA, NP-SC-PLA and NP-HC-PLA, respectively.

  1. Figure 3. The authors should show the inner structure.

The authors were unable to perform TEM of the inner structure of the PLA NP as our University facilities do not offer the micro-tomming capability required to achieve those experiments. However, we have performed cryo-SEM experiments instead of the gelatin hydrogel network as well as the hydrogel network loaded with the PLA NPs to provide extra morphological evidence of the designed DDS.

  1. There are many hard-to-sentences. The written English of the manuscript should be considerably improved, and therefore, a revision by a native speaker is highly recommended.

The authors thank the comment of the reviewer and the long sentences within the manuscript have been shortened to improve the flow of the main text.

  1. Is the PLA/gelatin ratio affected the results? The authors should discuss the points.

The authors have selected the PLA/gelatin ratio to provide the designed DDS with adequate mechanical properties to store the PLA NPs as well as to possess the right viscosity to allow the local applicability by syringe injection. The gelatin concentration was found in previous work (54). In addition, the NP concentration was selected to prevent agglomeration and percolate the gel network as proved by the additional provided rheological experiments as well as possessing enough drug encapsulated for therapeutic treatments. However, future work is needed to optimise the higher amount of loaded PLA NPs.

  1. Why is the DOX selected? Is it possible to apply cancer therapy? Is the result different among the drugs?

The authors thank the comment of the reviewer and have added the justification of the Doxorubicin selection to the manuscript:

Abstract

“The nanoparticles generated were successfully formulated with two different antitumoral (doxorubicin and dasatinib) possessing different structures to prove the loading versatility of the drug delivery system.”

Main text

“Polymeric nanoparticles from the selected PLA derivatives (NP-PLA) were successfully formulated with two different antitumoral by the double emulsion method and their size and morphology were characterized by DLS (Table 2) and SEM (Figure 3). doxorubicin and dasatinib were selected to assess the loading capacity of the NP-PLA with two different model compounds that possess distinct structures either featuring a stiff structure imposed by the aromatic groups for the doxorubicin or a flexible chain structure for dasatinib.”

Conclusion

“Furthermore, the multiprofile drug release obtained from the different components of the bionanocomposite could be apply for the treatment of complex tumours that need the administration of alternately drugs, such as for the urotherial carcinoma.”

  1. How about the degradation profiles of the nanocomposite particles?

The authors agree about the need of performing drug release kinetics experiments to enhance the scientific goal of the manuscript. We have performed as explained in question 4:

“The release kinetics of doxorubicin from the PLA NP as well as within the gelatin hydrogel network and finally from the PLANP within the hydrogel network has been monitored. The results have been compared to strengthen the applications of the designed DDS system.”

  1. Considering the inflammation induction or property control, PLGA must be appropriate material than PLA. Why is the PLA selected?

 Overall, the scaffold strength or novelty is not clear.

The authors acknowledge the view of the reviewer about the well-established properties of PLGA which will depend on the intended application anyway. Likewise, PLA could be preferred for the encapsulation of hydrophobic drugs due to its higher hydrophobic nature than PLGA. In addition, the paper does not disregard the future generation of PLGA derivatives by using our synthetic approach. In fact, the paper envisages future multiblock PLA derivatives using our catalyst that allow us to synthesise multiblock PLA as the presented herein di-block stereocomplex PLA derivative due to the living character of our polymerization. The paper stresses the influence of the synthesis, processing and application conditions on the final structure of the designed DDS and thus, the final release kinetics. Likewise, the novelty and strength of the presented bionanocomposite lie in the ability to design multi-drug delivery systems by selecting the most convenient PLA derivative with a tailored crystallinity which is hampered due to the NPs processing conditions and the low crystallinity rate featured by PLA. Here, we have presented a straight way to customise the crystallinity by the post-formulation annealing within the gelatin hydrogel network to avoid the NPs agglomeration whiles adding mechanical properties for both its storage and local application as well as making it more hydrophilic. Besides, the copolymerization of PLA with PGA produces generally a decrease in crystallinity or a fully amorphous polymer that limits the possibility of obtaining different release profiles related to its nanostructure as mentioned in the introduction. In contrast, PLA is a semicrystalline polymer that has several polymorphisms exhibiting different physicochemical properties. Particularly, the stereocomplex phase exhibits a longer degradation rate due to the stabilization of the crystal structure by hydrogen bonds, which could lead to a prolonged drug release. The stereocomplex phase is typically obtained from the blend of PLLA and PDLA due to the ease of production. However, the stereocomplexation of the PLA nanoparticles requires a long time from the solution (up to 7 days), as explained in the introduction. Therefore, the methodology presented in the manuscript allows not only the crystallization of the PLA nanoparticles without their aggregation but also, the formulation of PLA nanoparticles exhibiting stereocomplex phase within minutes instead of days is itself a novelty.

Round 2

Reviewer 1 Report

Author revised the manuscript title "Multifunctional PLA/Gelatin bionanocomposites for tailored drug delivery systems" and performed studies upto the drug release. In my opinion revised manuscript is acceptable.

Author should add some results in the abstract for reader point of view.

Overall this manuscript may be acceptable

Reviewer 3 Report

I think the author's comments to Reviewers 2 and 3 are reversed. Therefore, I clicked Reviewer 2 button and checked the corresponding comments.

This manuscript has been improved well. I recommend the publication.